# Coherent optical communications using coherence-cloned Kerr soliton microcombs

Yong Geng[1,3], Heng Zhou [1,3 ✉], Xinjie Han[1], Wenwen Cui[1], Qiang Zhang[1], Boyuan Liu[1], Guangwei Deng[2], Qiang Zhou [2] & Kun Qiu[1]

Dissipative Kerr soliton microcombs have been recognized as a promising multi-wavelength laser source for fiber optical communications, as their comb lines possess frequency and phase stability far beyond the independent lasers. Especially, for coherent optical communications, a highly beneficial but rarely explored target is to re-generate a Kerr soliton microcomb as the receiver local oscillators that conserve the frequency and phase property of the incoming data carriers, so that to enable coherent detection with minimized optical and electrical compensations. Here, via pump laser conveying and two-point locking, we implement re-generation of a Kerr soliton microcomb that faithfully clones the frequency and phase of another microcomb sent from 50 km away. Moreover, by using the coherence-cloned soliton microcombs as carriers and local oscillators, we demonstrate terabit coherent data interconnect, wherein traditional digital processes for frequency offset estimation are totally dispensed with, and carrier phase estimation is substantially simplified via slowed-down estimation rate per channel and joint estimation among multiple channels. Our work reveals that, in addition to providing a multitude of laser tones, regulating the frequency and phase of Kerr soliton microcombs among transmitters and receivers can significantly improve optical coherent communication in terms of performance, power consumption, and simplicity.

[1] Key Lab of Optical Fiber Sensing and Communication Networks, University of Electronic Science and Technology of China, Chengdu 611731, China. [2] Institute of Fundamental and Frontier Sciences, University of Electronic Science and Technology of China, Chengdu 611731, China. [3] These authors contributed equally: Yong Geng, Heng Zhou. ✉email: zhouheng@uestc.edu.cn

avelength division multiplexing (WDM) optical coherent transmission greatly enhances the capacity and spectral efficiency of fiber communication by modulating information onto both the amplitudes and phases of a multitude of laser carriers at the transmitter, and demodulating information at the receiver through coherently mixing the data signals with matched local oscillators (LO)[1]. Frequency and phase coherence between the carrier and LO lasers thus play a crucial role in determining the performance of coherent data receiving. To date, most commercial systems still use independent carrier and LO

lasers, which have weak mutual coherence (Fig. 1a, left) and entail large guard band and power-hungry digital signal processing (DSP) to gauge their frequency and phase uncertainties[2]. Optical frequency comb consisting of a large quantity of even spaced and phase-locked laser tones can provide spectral stability orders of magnitude higher than individual lasers (Fig. 1a, right)[3], thus being considered as a promising laser source for coherent WDM transmission. The strengths of optical frequency comb to carry massive parallel data channels have already been demonstrated in various of platforms, including electro-optical (EO) modulating comb[4–6], nonlinear

**Fig. 1 Coherence-cloned re-generation of DKS microcomb. a** Left: for conventional WDM system based on independent lasers, guard intervals are necessary to tolerate the random frequency drifts among adjacent channels at the transmitter side, while at the receiver side power-hungry DSP must be implemented to recover and compensate the random frequency and phase drifts between the carriers and LOs. Right: optical frequency comb has much better stability than independent lasers, thus holds great potentials to improve the spectral efficiency by eliminating guard intervals and reduce the DSP complexity of a coherent receiver. **b** Upper: schematics of coherence-cloned DKS microcomb re-generation and two-point locking. Lower: optical spectra of the original transmitter comb $C_{Tx}$ and re-generated receiver comb $C_{Rx}$. **c** Inter-comb beat note spectra between $C_{Tx}(m)$ and $C_{Rx}(m)$, $m = 1, 5, 10, 17$. Without two-point locking means that the conveyed pump laser $C_{Tx}(0)$ is used to generate the receiver microcomb $C_{Rx}$, but $C_{Rx}(17)$ is not locked to the pilot tone $C_{Tx}(17)$. **d** Allan deviations of the inter-comb beat note frequency $\Delta f_{CL}(m)$, $m = 1, 5, 17$, which confirm the efficacy of frequency stability enhancement by two-point locking. The fundamental repetition rate offset between $C_{Tx}$ and $C_{Rx}$ is $\Delta f_{CL}(1) = 55.358882$MHz.

broadened comb[7,8], mode-locked fiber laser comb[9], semiconductor gain-switched laser comb[10], and dissipative Kerr soliton (DKS) microcomb[11–16]. Therein, DKS microcombs generated in nonlinear optical microcavity have evoked special interests thanks to their unique features including large frequency spacing[11,12], ultra-broadband spectrum[17], high stability[18], excellent SWaP (size, weight and power) factors and compatibility with chip integration[19–21]. It had been reported that chip-scale DKS microcombs can simultaneously provide more than 100 laser tones to transmit coherent data signals with line rate up to 55 Tbit/s[12].

On the other hand, to employ DKS microcombs in coherent communication networks, it is of great importance to re-generate a LO microcomb at the receiver side that clones the frequency and phase coherence of the transmitted data carrier comb[22–24]. In fact, the generation dynamics and physical characteristics of DKS microcomb make it an ideal platform to realize coherence-cloned comb re-generation between the transmitter and the receiver[11,17,23,24]. First, DKS microcomb is commonly generated by a single continuous-wave pump laser, which directly sets the central frequency $f_c$ of the whole comb spectrum[11]. Second, the mode spacing $f_{spc}$ of a DKS microcomb is preset by the cavity geometry and can be finely adjusted by configuring either the intracavity pump power or the pump-cavity frequency detuning $\delta$ (i.e., fundamentally through changing the phase-matching condition)[23,25]. Third, in soliton mode-locked state, the phases of all the DKS microcomb lines $\phi_m$ ($m = 1, 2, 3,...$) uniformly align to the phase of the pump laser $\phi_0$[11,26]. That is to say, all the spectral parameters ($f_c$, $f_{spc}$ and $\phi_m$) of a DKS microcomb can be precisely manipulated by controlling the pump laser power and frequency.

Here, we demonstrate coherence-cloned DKS microcomb re-generation by relaying the pump laser between a pair of transmitter and receiver separated by 50 km, with the assistance of a pilot tone to achieve further mode spacing stabilization and phase noise suppression between the original and re-generated microcombs, based upon the mechanisms of two-point locking and optical frequency division. We illustrate that the re-generated receiver microcomb achieves excellent frequency and phase consistency with the transmitter microcomb, enabling high performance and energy-saving coherent data receiving with substantially simplified processing for the carrier-LO frequency offsets and phase drifts.

## Results

**Coherence-cloned re-generation of DKS microcomb.** Our experiment utilizes two silicon nitride micro-ring cavities with similar free spectral range (FSR) of ~100 GHz[27]. A low-noise 1-kHz linewidth fiber laser with wavelength $\lambda_{pump} \sim 1550.0$ nm is used as the pump laser field $C_{Tx}(0)$ to produce a DKS microcomb $C_{Tx}(m)$, $m = \pm1, 2, 3...$ in the transmitter microcavity, via the technique of auxiliary laser heating (ALH) (see Methods)[28–30]. ALH is adopted in order to suppress the thermal nonlinearity of microcavity resonances and allow the pump laser to stably access single soliton state in the red-detuning regime[29]. Afterwards, the transmitter microcomb $C_{Tx}$ and the pump laser $C_{Tx}(0)$ are sent through 50 km standard-single-mode fiber (SSMF) to the receiver, where the conveyed pump laser $C_{Tx}(0)$ is used to re-generate another DKS microcomb $C_{Rx}(m)$, $m = \pm1, 2, 3...$ in the receiver microcavity, also using ALH method[29]. Optical spectra of the transmitter microcomb $C_{Tx}$ and receiver microcomb $C_{Rx}$ are shown in Fig. 1b. Once generated, these microcombs can operate for weeks maintained by simple stabilization techniques[27]. At this stage, $C_{Tx}$ and $C_{Rx}$ can be considered as being 'one-point locked': both microcombs are generated by the same pump laser and thus have an identical central wavelength $\lambda_{pump}$, and the comb line phases $\phi_m^{Rx}$ and $\phi_m^{Tx}$, ($m = \pm1, 2, 3...$) all align approximately to the corresponding pump laser phase $\phi_0^{Rx}$ and

$\phi_0^{Tx}$ respectively (see detailed theoretical analysis in Supplementary Note 1)[26]. However, due to the distinct soliton repetition rates ($f_{spc}^{Tx} \sim 100.53$ GHz, $f_{spc}^{Rx} \sim 100.58$ GHz) and their uncorrelated jitters caused by the fluctuations of the two independent microcavities, the frequency and phase coherence between $C_{Tx}$ and $C_{Rx}$ are still weak[25,31], exhibiting inter-comb beat note spectrum with full-width-half-maximum (FWHM) linewidth >3 kHz (e.g., $m = 1$, as shown in Fig. 1c). Therefore, at this stage $C_{Rx}$ is not yet a coherence-cloned copy of $C_{Tx}$.

Then, we implement phase locking of the 17th receiver comb line $C_{Rx}(17)$ to the arrived 17th transmitter comb line $C_{Tx}(17)$ using an optical phase-lock loop (OPLL, see Methods), and narrow their beat note FWHM linewidth down to ~5 Hz (see Fig. 1c). By doing this, $C_{Tx}$ and $C_{Rx}$ are considered being two-point locked[32] by the shared pump laser $C_{Tx}(0)$, and the 17th comb modes $C_{Tx}(17)$ and $C_{Rx}(17)$, thus the inter-comb beat note phase noise of those in-between comb lines $C_{Tx}(m)$ and $C_{Rx}(m)$, ($m = \pm1, \pm2, \pm3, ... \pm16$) can be substantially suppressed obeying the law of optical frequency division (OFD)[33–35]. As shown in Fig. 1c, after two-point locking, the linewidths of inter-comb beat notes ($m = 1, 5, 10$) significantly decrease and their noise backgrounds drop approximately as the function of $1/(m - 17)^2$ (detailed phase analysis between $C_{Tx}$ and $C_{Rx}$ under the influences of fiber chromatic dispersion and random fiber length fluctuation can be found in Supplementary Note 1)[32,34]. Moreover, Fig. 1d shows the Allan deviation of the beat note frequency $\triangle f_{CL}(m)$ between $C_{Tx}(m)$ and $C_{Rx}(m)$, ($m = 1, 5, 7$), it is seen that two-point locking improves the stability of $\Delta f_{CL}$ by about 4 orders of magnitude (at 1 s gate time) comparing with the situation without two-point locking. Of note, here we choose $m = 17$ as the locked comb mode due to the bandwidth limitation (< 1.0 GHz) of our phase comparator. According to the theory of OFD, the locked comb mode index should be further increased by adopting either faster phase comparator or smaller discrepancy between $f_{spc}^{Tx}$ and $f_{spc}^{Rx}$, so that to obtain larger division factor and stronger coherence enhancement between $C_{Rx}$ and $C_{Tx}$. Nevertheless, the low noise beat notes and stable Allan deviations shown in Fig. 1c, d indicate that the mutual coherence between the DKS microcombs $C_{Rx}$ and $C_{Tx}$ is already high. Next, we will show how these highly coherent microcombs facilitate coherent data transmission.

**Coherent data interconnect using coherence-cloned microcombs.** Figure 2a illustrates the experimental setup of optical data interconnect using the coherence-cloned DKS microcombs as carriers and LOs[28]. At the transmitter, 20 comb lines $C_{Tx}(m)$ ($m = \pm2, \pm3,..., \pm11$) are selected using a wavelength selective switch (WSS) and sent into an IQ modulator (Fig. 2c), where 21 Gbaud s$^{-1}$ single polarization 16-QAM data are encoded on all the comb lines. The modulated data channels together with the pump laser $C_{Tx}(0)$ and pilot tone $C_{Tx}(17)$ are combined using an optical coupler and sent to the receiver through 50 km SSMF, the optical power launched into the fiber is −10 dBm for each channel, and the signal-to-noise ratio (SNR) is bigger than 25 dB (measured with a resolution bandwidth of 20 pm). At the receiver, the microcomb $C_{Rx}$ is re-generated and two-point locked to $C_{Tx}$ following the process described above, and then used as an array of LO for coherent date receiving (see Fig. 2c). Of note, as $C_{Tx}(0)$ and $C_{Tx}(17)$ propagate through the 50 km fiber link together with the high speed data, a matter of concern is that the data signals may impose linewidth broadening to them via nonlinear cross-phase modulation (XPM) and degrade their spectral purity as the pump laser and reference pilot. Nevertheless, as shown in Fig. 2d, e, after co-propagating with the data channels,

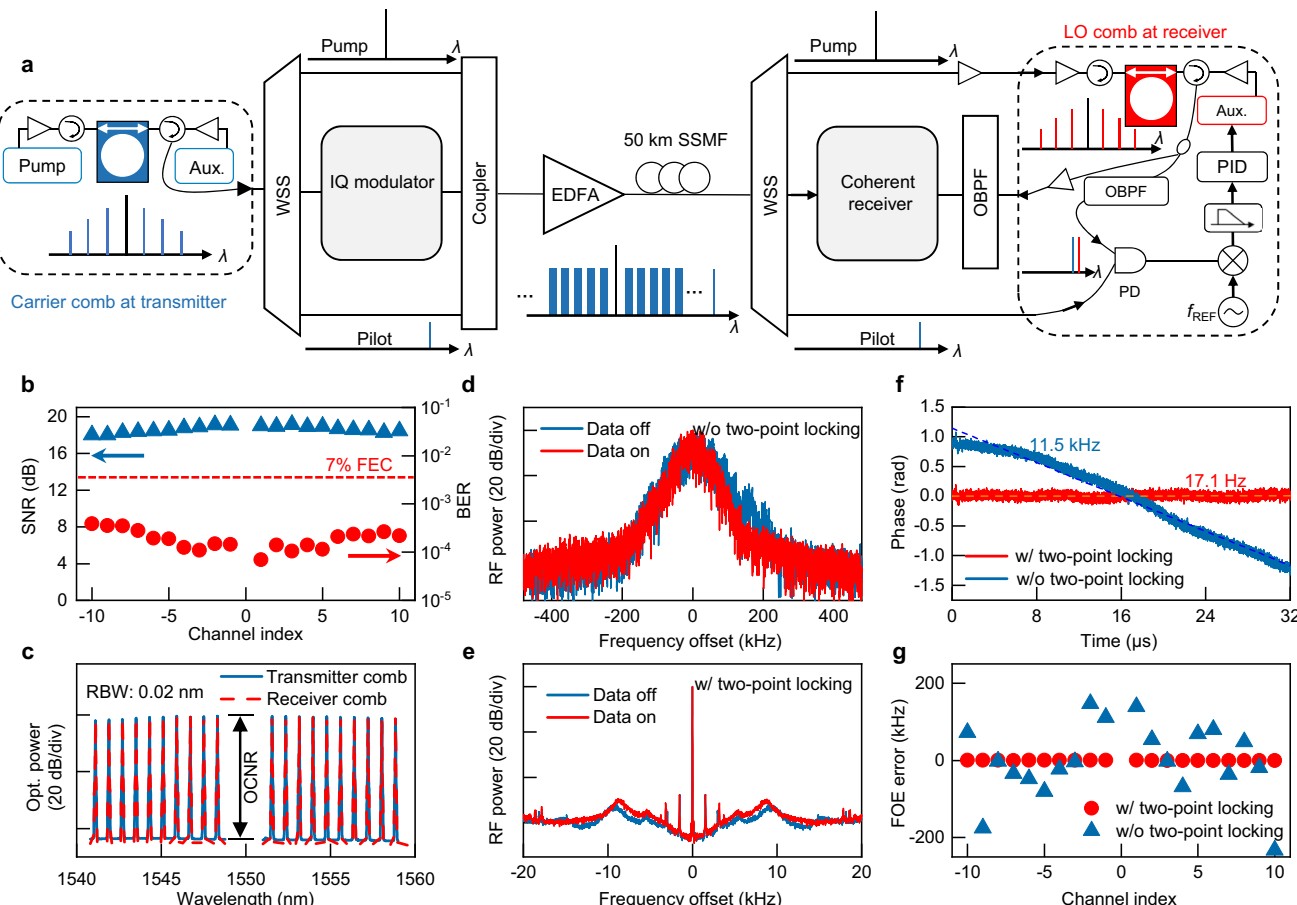

**Fig. 2 Optical interconnect using coherence-cloned DKS microcombs as carriers and LOs. a** Setup of the optical coherent data interconnect experiment. At the transmitter, microcomb $C_{Tx}$ is generated in a silicon nitride micro-ring cavity based on the ALH method and used as multi-wavelength laser carriers. A programmable wavelength selective switch (WSS) is adopted to select 20 comb lines from $C_{Tx}$ and sent them to a high speed IQ modulator, where 21 Gbaud 16-QAM data is encoded to all the comb lines, forming 20 data channels with total bitrate 1.68 Tbit s$^{-1}$. Although we used a common IQ modulator for all the channels, each comb line can carry independent data information encoded by different transmitters. The pump laser $C_{Tx}(0)$, pilot-tone $C_{Tx}(17)$ and 20 data channels are then coupled together and sent to the receiver through 50 km SSMF. At the receiver, microcomb $C_{Rx}$ is re-generated and two-point locked to $C_{Tx}$, then used as LOs for coherent data receiving. PID: proportional integral derivative; PD: photodiode; OBPF: optical bandpass filter. **b** SNR and BER measurements for all the 20 data channels, using two-point locked microcomb lines as carriers and LOs. **c** Optical spectra of the carrier and LO comb lines, showing high OCNR > 40 dB for each line. **d,e** Comparison of beat note spectra between $C_{Tx}(17)$ and $C_{Rx}(17)$ with and without two-point locking. **f** The resolved phase evolution of channel 1 when FOE is directly calculated via $m \cdot f_{REF}/17$. The dotted lines show the residual FOE error extracted from the first-order derivative of time from the actual data phase evolution, showing a 17.1 Hz (11.5 kHz) FOE error with two-point locking (without two-point locking). **g** Summarized FOE errors with and without two-point locking for all the 20 data channels.

the beat note linewidth between $C_{Tx}$ (17) and $C_{Rx}$ (17) remains almost identical with the case without co-propagating data, indicating that XPM only induces negligible linewidth distortion to $C_{Tx}$ (17) and $C_{Tx}$ (0) in our experiment. The underlying mechanism is because fiber dispersion induces spatiotemporal walk-off among signals at different wavelength channels along the transmission link[36,37], therefore, XPM imposed to $C_{Tx}(17)$ and $C_{Tx}(0)$ from different data channels are smoothed out as a quasi-constant phase envelop without high frequency component (relevant numerical simulation can be found in Supplementary Note 2).

Figure 2b shows the performance of coherent data receiving enabled by $C_{Tx}$ and $C_{Rx}$, it is seen that excellent SNR and bit-error ratio (BER) are achieved for all the 20 channels, with a total bit rate of 1.68 Tbit/s. More importantly, thanks to the excellent coherence between the two-point locked $C_{Tx}$ and $C_{Rx}$, DSP-based electrical frequency offset estimation (FOE) and carrier phase estimation (CPE) can be substantially simplified during coherent data retrieval[1,2,38,39].

First, after two-point locking, the frequency offset $\triangle f_{CL}(17)$ between $C_{Rx}(17)$ and $C_{Tx}(17)$ has been locked to the OPLL reference clock $f_{REF} = 941.101000 MHz$, with tiny residual frequency jitter < 1 Hz (at 100 ms gate time, see Fig. 1d), so the FOE between $C_{Tx}(m)$ and $C_{Rx}(m)$, $(m = \pm 1, 2,...,16)$ can be precalculated using the simple relation $\triangle f_{CL}(m) = m \cdot f_{ref}/17$. To validate this scheme, we conduct coherent data demodulation using the precalculated $\triangle f_{CL}(m)$ as the FOE result for each channel, and resolve the actual frequency offset error by extracting the first-order derivative of time from the data phase evolution (see Fig. 2f)[37]. As summarized in Fig. 2g, the discrepancies between the precalculated $\triangle f_{CL}(m)$ and actual frequency offsets are within ± 500 Hz for all the 20 data channels. In comparison, when $C_{Rx}$ and $C_{Tx}$ are not two-point locked but only one-point locked, the errors of the precalculated $\triangle f_{CL}(m)$ become ~3 orders of magnitude bigger. Hypothetically, if traditional DSP algorithms for FOE (e.g., 4th order fast Fourier transformation) are used to achieve such accuracy of ±500 Hz, it

would entail unacceptably heavy DSP operations and super-long training sequence (see quantitative analysis in Supplementary Note 3)[38]. Thus, by virtue of the cloned frequency stability between $C_{Tx}$ and $C_{Rx}$, one can save substantial DSP power and complexity while simultaneously achieving FOE accuracy that is orders of magnitude higher than relying on conventional digital methods.

Second, besides FOE, to retrieve data information from coherently modulated signal, random phase drift between the carrier and LO needs to be traced using CPE algorithms[1,38]. Essentially, the processing rate and corresponding power consumption of CPE depend on the phase coherence between the carrier and LO tones[39]. In other words, the lowest CPE rate should be properly chosen to minimize the power consumption (considering that typical CPE algorithms such as blind-phase search are usually power hungry, see quantitative analysis in Supplementary Note 3), while making sure that stochastic phase drift within the interval between two CPE operations causes acceptable bit errors[38,39]. It has been shown above that the phase coherence between $C_{Rx}$ and $C_{Tx}$ is greatly enhanced by two-point locking and OFD, so it is expected that the CPE rate and related power consumption can be reduced when using them as carriers and LOs. As shown in Fig. 3b, the data channels carried and demodulated by $C_{Rx}$ and $C_{Tx}$ exhibit stable phase evolution (see Methods for phase retrieval scheme), with the phase fluctuation much smaller than generated by one-point locked microcombs. Larger indexed data channels show slightly bigger phase fluctuations, because the corresponding comb lines experienced smaller frequency division factors, but the phase fluctuations are still confined in a small range (e.g., $< \pm 0.2$ rad). Then we gradually slow down the CPE rate (i.e., increasing the number of skipped data blocks after each CPE) while continuously recording the BER and evaluating the lowest CPE rate allowed for different data channels. As shown in Fig. 3c, if we set the 7% hard forward-error-correction (FEC) threshold 3.8e−3 as the target BER, two-point locked $C_{Rx}$ and $C_{Tx}$ enable 1 order of magnitude lower CPE rate than one-point locked microcombs, and 3 orders of magnitude lower CPE rate than free-running independent carrier and LO lasers. Particularly, for those lower indexed data channels (e.g., $m < 5$), only one CPE block (32 symbols) is sufficient to warrant satisfying BER of the whole data frame (400,000 symbols), implying substantial power saving of the relevant DSP module. Practically, such stable phase evolutions between the coherence-cloned $C_{Rx}$ and $C_{Tx}$ can even be tracked by adaptive equalizer module without conducting CPE (as demonstrated in Supplementary Note 3). Detailed hardware and algorithm design of a coherent receiver that fully copes with the coherence-cloned microcombs is beyond the scope of the present work, but will be an important topic as Kerr microcomb moving fast towards utility.

Furthermore, Fig. 3e shows the data phase evolutions of different channels that are simultaneously demodulated in two coherent receivers, it is observed that strong phase correlations exist among different channels. Such phenomenon is based on the fact that, in our 50 km interconnect experiments, the fast phase fluctuations between the two-point locked microcombs primarily result from the residual phase noise of the OPLL: $\frac{m}{17}\phi_{OPLL}(t)$ (see Supplementary Eq. S12 and Supplementary Fig. 1a), which linearly scales up with $m$ and means that we can use the CPE result of one channel to predict the phases of the other channels in a master-slave fashion, as sketched in Fig. 3d[5,37]. Figure 3f shows the measured data receiving performance when master-slave CPE is conducted among channel 1 to channel 10, excellent BER is achieved when the phase of higher indexed data channel (e.g., channel 10) is used to detect lower indexed data channels

(e.g., channel 1 to 9). For example, when the CPE result of channel 10 is used for channel 1, only minor BER penalty is observed in comparison with the result of independent CPE. Moreover, comparing with recent demonstrated master-slave CPE using uncorrelated carrier comb and LO comb[37], coherence-cloned DKS combs possess much longer mutual coherent time, so they should be less sensitive to phase de-coherence caused by fiber dispersion (see Supplementary Note 1), and thus can potentially support the master-slave CPE strategy in longer transmission distance scenarios.

## Discussion

Synthesizing the results in Fig. 3c, e, we can choose a desired trade-off between BER performance and CPE simplicity according to specific system requirements. For example, if our system has a target BER of 3.8e−3, we can run CPE every other 1001 data block (i.e., skipping 1000 blocks after each CPE) for channel 10 and use the CPE result to detect channel 9 to channel 1. So, only 13 CPE operations ($\lceil 12,500 \div 1000 \rceil$) are needed for channel 10 and the total decoded symbols sums up to 4,000,000 (400,000 symbols per channel × 10 channels). In comparison, if independent carriers and LOs are used, 12,500 CPE operations (12,500 ÷ 10 × 10 channels) are needed to reach the 3.8e−3 BER (i.e., skipping 10 blocks after each CPE, see Fig. 3c black curve) within 4,000,000 symbols. According to such evaluation, 10 data channels carried and detected by coherence-cloned microcombs ($i = 1000$, $j = 9$, the meanings of $i$ and $j$ are exhibited in Fig. 3a, d) bring about more than 3 orders of magnitude less CPE operations comparing with the same data capacity implemented by individual laser carriers and LOs ($i = 10$, $j = 0$). Such prominent source saving can further scale up when the number of data channel increases. But then again, to optimize the performance of a practical coherent receiver, it needs comprehensive design of multiple hardware and software modules (analogue to digital conversion, timing recovery, equalization, FOE, CPE, and decoding), how to maximize the merits of coherence-cloned microcombs to facilitate the collaborative operation of these modules entails further exploration, both on the scientific and engineering aspects.

In summary, we have demonstrated coherence-cloned re-generation of DKS microcombs over long distance and used them as the transmitter carriers and receiver LOs for terabit coherent data interconnect. Enabled by the schemes of two-point locking and OFD, excellent frequency and phase coherence are achieved between the original and re-generated microcombs, which are leveraged to realize totally saving of FOE and substantially reducing of CPE in coherent data detection. Our experiment provides a potential solution to improve the spectral regularity of WDM system from GHz to sub-kHz, using integrated photonic devices. Indeed, our experiment only demonstrated simple point-to-point interconnect, for which the pump laser and pilot tone can be conveniently conveyed from the transmitter to the receiver. Such scheme would become challenging for networks with multiple nodes and sophisticated topology[13]. However, instead of conveying the pilot tones, two-point locking among the transmitter and receiver microcombs can also be implemented using local optical frequency standards such as atomic gas cells or ultra-stable optical cavities[40]. As long as the mutual frequency and phase stability among microcombs at different network nodes are sufficiently high, the above-demonstrated benefits regarding FOE and CPE for coherent data receiving can be obtained, offering a potential solution to cope with the impending energy crisis that vexes the optical transmission industry.

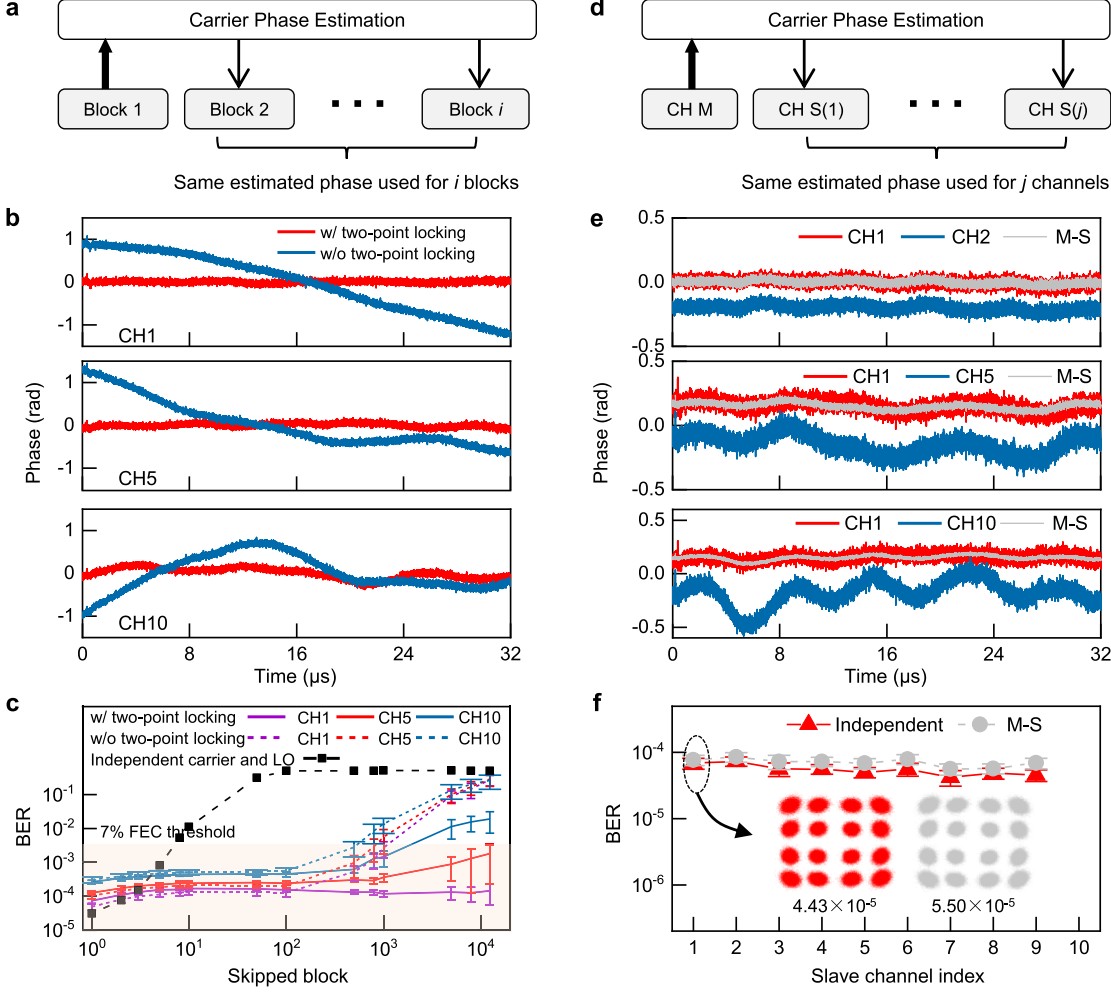

**Fig. 3 Carrier phase estimation facilitated by coherence-cloned DKS microcombs. a** Scheme for CPE rate configuration. CPE is conducted once every $(i + 1)$ data blocks, namely, the CPE results of block 1 is used for the following $i$ data blocks. **b** CPE results for channel 1, 5, and 10. It is seen that two-point locking between $C_{Tx}$ and $C_{Rx}$ significantly enhances the phase stability between carriers and LOs. For this measurement the data format is 12.5 Gbaud 16-QAM, each panel illustrates a time window of 32 μs containing 400,000 symbols. **c** Measured BER as a function of CPE rate. The block size for pilot-based CPE is 32 symbols. It is obvious that coherence-cloned microcombs allow much slower CPE rate to reach the target BER 3.8e−3 comparing with unlocked microcombs and independent carriers and LOs. The error bars show the standard deviation (S.D.) of 15 BER measurements at each number of skipped block. **d** Scheme for master-slave joint CPE among multiple data channels. The carrier phase is estimated from the master channel (CH M) and then applied to $j$ slave channels (CH S). **e** Retrieved data phases by individual CPE and master-slave CPE. Upper: channel 1 as slave and channel 2 as master; Middle: channel 1 as slave and channel 5 as master; Lower: channel 1 as slave and channel 10 as master. **f** Summarized BER performance of individual CPE and master-slave CPE of multiple data channels. For master-slave CPE measurement, channel 10 is used as the master channel and channel 1 to 9 are processed as slave channels. The inset shows the constellation maps and BER for channel 1 retrieved via individual and master-salve CPE.

## Methods

**Generation and locking of $C_{Tx}$ and $C_{Rx}$.** DKS microcomb $C_{Tx}$ is first generated in the transmitter microcavity, using the auxiliary laser heating method[29,30]. Particularly, an auxiliary laser is tuned into the blue-detuning regime of a cavity mode (~1536.2 nm), and subsequently a pump laser is tuned into another cavity mode (~1549.9 nm). By properly setting the power and detuning of the pump and auxiliary laser, the heat flow caused by them can be balanced out, allowing the pump laser to stably scan into the red-detuning regime and access single soliton state[29]. Moreover, using the same pump laser sent from the transmitter to the receiver through 50 km SSMF, DKS microcomb $C_{Rx}$ is similarly re-generated in the receiver microcavity, by using another auxiliary laser (~1536.2 nm) to simultaneously control the pump detuning and maintain the cavity thermal stability[27].

To achieve two-point locking, $C_{Tx}(17)$ and $C_{Rx}(17)$ are filtered out and sent into a fast photodiode in which their beating frequency $\triangle f_{CL}(17)$ is detected. Then, $\triangle f_{CL}(17)$ and a reference clock $f_{REF} = 941.101000$ MHz are sent into a phase comparator to generate the error signal and feedback control the power of the auxiliary laser for generating $C_{Rx}$. Particularly, the auxiliary laser power controls the pump detuning in the receiver microcavity via thermal resonance shift, and in turn adjust the repetition rate of $C_{Rx}$[25,27], so that to lock $C_{Rx}(17)$ to $C_{Tx}(17)$. The bandwidth of our phase lock loop is about 100 kHz, set by the amplitude modulation frequency limitation of the adopted auxiliary laser module.

**Coherent data modulation and receiving using $C_{Tx}$ and $C_{Rx}$.** At the transmitter, 20 comb lines $C_{Tx}(m)$ ($m = \pm 2, \pm 3, ..., \pm 11$) are filtered out by a C-band programmable WSS and used as the data carriers. Each of the 20 comb lines is boosted to about 0 dBm using a low-noise Er-doped fiber amplifier (EDFA) while maintaining >40 dB optical carrier-to-noise-ratio (OCNR)(Fig. 2c). An IQ modulator is used to encode single-polarization 16-QAM data onto all the 20 comb lines, driven by an electrical arbitrary waveform generator (eAWG) to generate the 16-QAM waveform with rectangle pulse shaping. After modulation, all the data channels are amplified by another EDFA to generate −10.0 dBm launched power for each channel.

The 20 data channels together with the pump laser $C_{Tx}(0)$ and pilot tone $C_{Tx}(17)$ are transmitted through 50 km SSMF to the receiver. At the receiver, $C_{Rx}$ is re-generated as LOs for coherent data detection. Each data channel and the corresponding LO is selected by tunable optical band pass filters (OBPF) and fed into a coherent receiver. The detected electrical signal of each channel is recorded by a real-time digital phosphor oscilloscope (DPO) and then processed offline. Multiple algorithms are used to achieve optimal data retrieval[41]:

**Step 1:** The chromatic dispersion of the 50-km-long SSFM link is compensated in a fixed frequency-domain equalizer (FDE). And the signal is re-sampled to twice the symbol rate.

**Step 2:** The frequency offset between the carrier and LO of each channel is calculated and compensated by using the simple relation $\triangle f_{CL}(m) = m \cdot f_{ref}/17$, without conducting traditional DSP-based FOE algorithm.

**Step 3:** The IQ imbalance of the received data is compensated based on Gram-Schmidt orthogonalization. This is followed by a synchronization and re-timing operation based on the square timing recovery algorithm.

**Step 4:** A Volterra equalizer is used to alleviate data distortion due to link impairments. In our 50 km interconnect experiment, the nonlinear distortions of the data signals are not prominent, so the Volterra equalizer mainly consists of 30 linear taps (and only two second-order taps and one third-order tap). Also, the equalizer is configured in a static fashion, i.e., the taps are trained using the first 2000 symbols of each data frame and then fixed for the rest symbols. Using static equalizer warrants that the phase variations between the transmitter and receiver comb lines are solely handled by the CPE module. More complicated adaptive equalizer is also used to detect the data collected in our experiment, which can simultaneously compensate the link impairments and the small residual phase errors between the microcombs, as demonstrated in Supplementary Note 3.

**Step 5:** Then, phase estimation and compensation is performed via pilot-aided CPE. The pilot symbols are 16-QAM data and each CPE process uses 32 symbols as the unit block size. Blind CPE can also be used to correctly retrieve the data, with similar performance as obtained via pilot-aided CPE (see Supplementary Note 3). Finally, the BER for each channel is obtained by comparing the received symbols with the originally sent symbols from the transmitter.

Besides, for the investigation of master-slave CPE process presented in Fig. 3d−f, two coherent receivers and four DPO channels are used to simultaneously receive two 16-QAM channels modulated at 12.5 Gbaud (i.e., 400,000 symbols within a 32 µs time window), so as to reserve the phase correlation between the two channels for master-slave phase estimation.

## Data availability

The raw data generated in this study have been deposited in Zenodo[41].

## Code availability

The codes supporting the findings of this study have been deposited in Zenodo[41].

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

## Acknowledgements

The authors thank Professor Chee Wei Wong for helpful comments and suggestions on this work, and VLC Photonics S. L. and LiGenTec SA for device fabrication. This work is supported by National Key Research and Development Program of China (2019YFB2203103, 2018YFA0307400); National Natural Science Foundation of China (62001086, 61705033, 61775025); The 111 project (B14039).

## Author contributions

The experiments were conceived and designed by Y.G., H.Z., and K.Q. Measurements were performed by Y.G., H.Z., X.H., and W.C., with assistance from Q.Z. and B.L. G.D. and Q.Z. built the OPLL. All authors analyzed the data and prepared the manuscript. Q.Z. and K.Q. supervised the project.

## Competing interests

The authors declare no competing interests.

**Additional information**

