## [Peer Review File · Nature Communications]

REVIEWER COMMENTS

Reviewer #1 (Remarks to the Author):

Comments on "Coherent optical..." by Geng et al.

This paper describes a transmission experiment based on frequency combs. The main novelty is that the authors have phase-locked two separate microcombs to each other, that are then used as signal and LO in a WDM-type transmission system. In principle this idea is not novel and was demonstrated previously with electro-optic combs but it is much more difficult with microcombs whose rep rate cannot be tuned with the same ease.

Therefore I am positive in general to this work being published, but there are many details that can and should be improved before publication. I will outline them below.

1) The language is understandable but can be improved in many aspects, e.g. the choice of articles and lack of determiners for singular nouns should be checked and improved throughout. For example the first sentence in the abstract: "Dissipative Kerr soliton microcomb has been recognized as.." This would be better as "Dissipative Kerr soliton microcombs have been recognized as..". There are numerous issues like this throughout, and the manuscript would greatly benefit from an overview by a native or at least a very good English speaker.

Also please check the ref list. Many titles are not properly capitalized (e.g. ref 2: dsp -> DSP)

2) I think fig 2a is highly misleading in that it indicates different modulators are used for the data, when in fact a common modulator is used for all channels (except the unmodulated $m=1$ and 17). I assume a waveshaper or similar was used for the demultiplexing, or how was it done? Also the mux in the TX - was another waveshaper used or what component was used for that? What was the B2B SNR after the final EDFA, and what was the power per channel into the fiber? Was the waveshaper(s) used to equalize the power between the channels since the input comb is by no means flat?

3) Some more details on the comb lines used for modulation and pilot-tones need to be specified. In the text they say $m=0$ and 17 are used, but the spectrum in 2c seems to indicate that channels like $m=-15$ to -5 and $m=5$ to 15 are used for data. Also lines $m=0$ and $m=-17$ is indicated in the figs rather than $+17$ (assuming the enumeration is increasing from left to right with 0 in the center).

4) Related to 3): When looking at the combs in fig 1b it seems like the center part has two dips, one at 1550 (where it is pumped, or..? - and by the way where is the pump located?) and one dip at 1535 or so - why is that? I would like to see a zoomed-in spectrum of the comb lines used in the transmission experiment before the demux, e.g. like in 2c but including also the specific pilot tones and possibly intermediate tones.

5) Another issue I have is the dispersive walkoff between the two pilot tones used to seed the Rx comb. In [36] this was discussed and shown to degrade the linewidth of the regenerated comb lines, and since the two seed lines are more widely separated in this experiment than in [36] the dispersive walkoff is larger and potentially more devastating for longer transmission distances. So my question is how the authors' scheme would scale with transmission distance. Apparently it works for 50 km, but would it work for $100/500/1000$ kms? This could be addressed in the supplementary.

6) I have numerous problems with the theory of comb phases in the "methods" section.

a) First of all the purpose of the calculation in eqs (1)-(10) is not clear. What do you want to show? Also I doubt this theory belongs to a methods section. Since this is an experimental paper I expect experimental details, microcomb details, setups and phase locking information, receiver DSP and details on how to measure and show the plotted quantities would appear here.

b) Clearly define the variables and the notation. It is not OK to refer to a specific phase value in time ϕ_{int}^{Tx} as is done after (1) as "intrinsic phase noise (i.e. linewidth)". Rather - I assume - all the ϕ 's are functions of time and a realization of a stochastically varying phase in time, whose statistical properties (which?) are a function of the laser's linewidth. Can you say anything about the statistical properties of these phases? Variance, ergodicity, pdf etc.? This applies also to the Δf_{Rep}^{Tx} which probably varies with time in some unknown fashion, or...?

c) The ϕ_{nl} contribution is neglected in the end and seem not to affect the calculation in any way, so I see no purpose in including it. It is also not clear how it would depend on m .

d) The authors seem to assume (given the account after eq 10) that the ϕ_{ff} (the phase change due to random fluctuations in the fiber) should also account for the dispersive delay? I think this is not correct and it is not how the delay is modeled. Instead if $\phi^0(t)$ is the phase of line 0 and $\phi^m(t)$ is the phase of line m in to the fiber, the phases after the fiber would be $\phi^0(t-T)$ and $\phi^m(t-T-\Delta t)$, where T is the group delay in the fiber and Δt is the dispersive delay (walkoff) between line 0 and line m . Assuming ergodicity, the common delay T can be neglected but the dispersive delay cannot. This walkoff is a serious problem that contribute to the net phase noise as discussed in Ref [36] and it should be seriously accounted for. The resulting phase noise varies on a time scale related to the laser coherence and not the fiber fluctuations. See also point 5) above.

e) Eq (5) seems to assume the variance of the RF contribution of the phase noise increase with line number - which I agree holds for electrooptic combs. However, how the phase noise scales with line number for microcombs is not clear as far as I know. If the authors want to use this model for microresonator-combs they need to support it with a reference or a separate measurement.

7) More details and explanations are needed on the coherent receiver DSP. I suggest providing a detailed explanation in a supplementary chapter. Specifically:
- the Volterra equalizer is nonstandard. What does it do? Why is it needed? What is the performance without it?
- is the received data pol muxed or single polarization, and what poldemux algorithm is used?
- apparently pilot-aided CPE is used. Details please - why is it needed if the LO is 'cloned' to the transmitter source? Note that this is a critical point because the whole purpose of this paper falls if the data cannot be retrieved with a blind CPE. Does it work without the CPE? What is the penalty without the CPE? What is the rep rate for the pilot symbols and what format are they?

8) You seem to compare the two-point locking to the free-running case, but what about 1-point locking - i.e. if you use line 0 to create the Rx comb and skip line 17 - is that case studied at all, and if so what is the performance?

These are some of the questions I would like to see addressed satisfactory before I would approve publication of this submission.

Reviewer #3 (Remarks to the Author):

The manuscript by Yong Geng et al. entitled "Coherent optical communications using coherence-cloned Kerr soliton microcombs" presents the experiments dedicated to coherence-cloned regeneration of DKS microcombs and their exploitation as the carriers and local oscillators for terabit coherent interconnect. The microcomb on the receiver side is regenerated by means of a microresonator which has a free spectral range similar to that of the transmitter, located 50 km away, by the conveyed pump that is transferred together with the channels for information transfer. In order to further increase the coherence between the microcombs, a two-point locking technique of the 17th comb lines is employed, so that beat note linewidth drops from > 3 kHz down to ~ 5 Hz. High coherence rate enables simpler and less power-consuming digital signal processing (DSP) for frequency offset estimation (FOE) and carrier phase estimation (CPE) to retrieve the information.

The manuscript is written well. The abstract contains a concise overview of the paper main results as well as their place in the framework of the considered field of research. Although the general distribution of the material between the main text and the supplementary information makes sense, the latter could be still enriched with explicit description of the conventional FOE and CPE algorithms that are said to be simplified.

The work is consistent with previous works on coherent data detection where the microcombs are used as a multiwavelength source that can be used instead of independent laser arrays to carry information on long distances and to be coherently retrieved on the receiver side. In our view, the introduction is properly composed so that it clearly describes the field and reasonably motivates the current research work. Although the coherent detection of the carrier microcomb by a local oscillator microcomb as well as regeneration of microcombs by a comb line pumping a remote microresonator were previously done in Ref. [MarinPalomo2017] and Ref. [Liao2019], correspondingly, the technique of two point locking, which principal foundations were described in Ref. [Fortier2011], was for the first time used to increase the coherence between the carrier and regenerated combs. On the other hand, as it has been demonstrated in Ref. [Jang2018], it is possible to generate a synchronized microcombs when the slave microcomb is locked to the master microcomb. Would it be possible to perform microcombs synchronization in the experiments considered in the manuscript instead of two-point-locking and pump conveying?

The issue of possible spectral purity degradation of the pump and pilot signals translated to the receiver due to the effect of nonlinear cross-phase modulation is addressed and found to be non-essential in the corresponding experimental comparison between the two cases when the both signals propagate together with the data carriers and without them. The explanation given to this observation is that fiber dispersion induces the spatiotemporal walk-off among signals at different wavelength channels and is confirmed by numeric simulations. Although the parameters set for these simulations are indicated, it could be more educational to include the basic equations comprising the numerical model and the method used for its simulations.

Rigorous analytic description of coherence analysis between the transmitter and receiver combs provides a clear insight into the origin of phase correlations between the channels observed in the experiment. This analysis provides a reasonable justification of the DSP algorithms covered in the paper. For more strict assertion of what advantages the proposed detection scheme offers, it would be useful to evaluate the asymptotic value of operations' number that are required by the conventional realizations of FOE and CPE and their upgraded counterparts.

In Fig. 2b, SNR and BER data are given, however, it is not clear which case these data correspond to: either it is for the two-point locked local oscillator (LO) comb or for the comb without it. Anyway, could you compare the both cases with the case when there is no cloning of the LO comb, please, to properly outline advantages offered by this technique?

In Fig. 3b, the phase evolution is given for the scale of tens of microseconds. At the same time, for the given data transmission rate, a time slot duration for one symbol transmission is on the order of tens of picoseconds. On this time-scale, the considered large-time-scale phase evolution does not contribute too much to the decoding error as the phase fluctuations are small. Why does one have to use two-point locking to increase the stability on the larger time scale, if that is the case?

In summary, the experiments is a system level experiment that demonstrates coherence cloning and is a very useful addition to the application of microcombs in coherent communication. The manuscript and experiments are of high quality.

Please find below a set of other questions and technical remarks which would improve the quality of the manuscript:

Line 135: BER stands for bit-error rate, however, in Ref [MarinPalomo2017] it stands for bit error ratio that is a more correct way to express the meaning of this quantity as it is dimensionless, whereas rate usually represents the speed of a physical measure evolution over time.

Line 281: ... linearly depends ...

Fig. 1b caption: ... must be implemented to recover ...

Fig. 2c: it is not clear how OCNR was evaluated. What was taken as the noise floor? Resolution bandwidth should also be added to properly represent the data collected from an optical spectrum analyzer.

Fig. 3c: BER should not be ranged up to 10. The ratios of more than 0.1 are meaningless to indicate.

Fig. 3f: why the shapes comprising the constellations are tilted? Why these shapes' distribution is non-uniform? (compare with Fig. 2g in Ref. [MarinPalomo2017])

Fig. S1b: ... between the models including and excluding the XPM effect ...

Fig. S2a: between OBPF and the circulator on the output there is a part of the setup scheme with the oval share which role is not clear.

References:

[MarinPalomo2017] Pablo Marin-Palomo, Juned N. Kemal, Maxim Karpov, Arne Kordts, Joerg Pfeifle, Martin H. P. Pfeiffer, Philipp Trocha, Stefan Wolf, Victor Brasch, Miles H. Anderson, Ralf Rosenberger, Kovendhan Vijayan, Wolfgang Freude, Tobias J. Kippenberg, and Christian Koos. Microresonator-based solitons for massively parallel coherent optical communications. *Nature*, 546 (7657) : 274–279, 2017

[Fortier2011] T. M. Fortier, M. S. Kirchner, F. Quinlan, J. Taylor, J. C. Bergquist, T. Rosenband, N. Lemke, A. Ludlow, Y. Jiang, C. W. Oates, and S. A. Diddams. Generation of ultrastable microwaves via optical frequency division. *Nature Photonics*, 5(7) : 425–429, 2011

[Jang2018] Jae K. Jang, Alexander Klenner, Xingchen Ji, Yoshitomo Okawachi, Michal Lipson, and Alexander L. Gaeta. Synchronization of coupled optical microresonators. *Nature Photonics*, 12(11):688–693, 2018.

[Liao2019] Peicheng Liao, Changjing Bao, Ahmed Almaiman, Arne Kordts, Maxim Karpov, Martin Hubert Peter Pfeiffer, Lin Zhang, Fatemeh Alishahi, Yinwen Cao, Kaiheng Zou, Ahmad Fallahpour, Ari N. Willner, Moshe Tur, Tobias J. Kippenberg, and Alan E. Willner. Demonstration of multiple Kerr-frequency-comb generation using different lines from another Kerr comb located up to 50 km away. *Journal of Lightwave Technology*, 37(2) : 579–584, 2019.

■ Reviewer1

Comments to the Author

Comment 0: Comments on "Coherent optical..." by Geng et al.

This paper describes a transmission experiment based on frequency combs. The main novelty is that the authors have phase-locked two separate microcombs to each other, that are then used as signal and LO in a WDM-type transmission system. In principle this idea is not novel and was demonstrated previously with electro-optic combs but it is much more difficult with microcombs whose rep rate cannot be tuned with the same ease.

Therefore I am positive in general to this work being published, but there are many details that can and should be improved before publication. I will outline them below.

Reply: We thank the reviewer for reviewing our manuscript and for being positive in general, the questions and suggestions really help us to improve our work. Below we have addressed your comments point-by-point. Thank you.

Comment 1: The language is understandable but can be improved in many aspects, e.g. the choice of articles and lack of determiners for singular nouns should be checked and improved throughout. For example the first sentence in the abstract: "Dissipative Kerr soliton microcomb has been recognized as.." This would be better as "Dissipative Kerr soliton microcombs have been recognized as..". There are numerous issues like this throughout, and the manuscript would greatly benefit from an overview by a native or at least a very good English speaker.

Also please check the ref list. Many titles are not properly capitalized (e.g. ref 2: dsp -> DSP)

Reply: As suggested by the reviewer, we have carefully revised our manuscript regarding the grammar and wording, based on advices from native English speakers.

Action taken: Revisions have been made throughout the manuscript to correct the grammar issues and improve expression. The revisions regarding language have been marked as red-text-with-yellow-highlight.

Comment 2: 2) I think fig 2a is highly misleading in that it indicates different modulators are used for the data, when in fact a common modulator is used for all channels (except the unmodulated $m=1$ and 17). I assume a waveshaper or similar was used for the demultiplexing, or how was it done? Also the mux in the TX - was another waveshaper used or what component was used for that? What was the B2B SNR after the final EDFA, and what was the power per channel into the fiber?

Was the waveshaper(s) used to equalize the power between the channels since the input comb is by no means flat

Reply: We thank the reviewer for the above questions.

First, in our experiment we indeed used a common IQ modulator for all the data channels, which

is a conventional measure for proof-of-concept experiment of multi-channel WDM transmissions, because in laboratory we usually do not have dozens of coherent transceivers. In the prior Fig. 2a we drew multiple IQ modulators to reveal that essentially different comb lines should be encoded with independent data. Such conceptual schematics were conventionally adopted in literature (e.g., Nature, vol. 546, no. 7657, 2017, pp. 274–279; Nature Communications, vol. 11, no. 1, 2020, p. 201; and Nature Communications, vol. 11, no. 1, 2020, p. 2568). Nevertheless, we agree with the reviewer that it is better to show the exact experiment setup to the readers, so we have modified Fig. 2a to include a common IQ modulator for all the channels, and clarified in Fig. 2a caption that “although we used a common IQ modulator for all the channels, each comb line can carry independent data information encoded by different transmitters”.

Second, yes, we used several wavelength selective switches (Finisar WaveShaper W400s, and Santec WSS-1000) and tunable optical bandpass filters (Santec OTF 350) in our experiment, to selectively multiplex and demultiplex the comb lines and data channels. Those WSS and OBPF modules have been added to the revised Fig. 2a and described in the Method subsection 2.

Third, the B2B SNR after the final EDFA was >25 dB (RBW: 0.02nm) for each of the 20 channels in our experiment, and the power per channel launched into the fiber was set to -10.0 dBm (+3.0 dBm for all the 20 channels). As suggested by the reviewer, these two values have been added to the revised manuscript.

Fourth, the comb powers shown in Fig. 2c were NOT equalized by the waveshaper. The central comb lines $m=\pm 11$ exhibited sufficiently flat powers (please see our answer to reviewer’s Comment 5) and becomes even flatter after being amplified by the EDFA (see Fig. 2c).

Action taken:

- 1) Fig. 2a has been modified to include a shared common transmitter for all data channels, and it has been clarified in Fig. 2a caption that “although we used a common modulator for all channels, each comb line can carry independent data information encoded by different transmitters”.
- 2) The wavelength selected switch (WSS) and tunable optical bandpass filter (OBPF) used in our experiment have been added to Fig. 2a, and described in the Method subsection 3.
- 3) The power and SNR per channel into the fiber have been added to the revised manuscript page 5 paragraph 2: “the optical power launched into the fiber is -10 dBm for each channel, and the SNR is bigger than 25 dB (measured with a resolution bandwidth of 20 pm).”

Comment 3: 3) Some more details on the comb lines used for modulation and pilot-tones need to be specified. In the text they say $m=0$ and 17 are used, but the spectrum in 2c seems to indicate that channels like $m=-15$ to -5 and $m=5$ to 15 are used for data. Also lines $m=0$ and $m=-17$ is indicated in the figs arther then $+17$ (assuming the enumeration is increasing from left to right with 0 in the center).

Reply: We thank the reviewer for this question.

First, as mentioned in the main text page 5 paragraph 2, the 20 comb lines used for data

transmission were indexed as $m=\pm 2, \dots, \pm 11$ (data channel indexed as CH 1 was actually carried by comb line $m=2$, and so on). The two comb lines adjacent to the pump laser (i.e., $m=\pm 1$) were abandoned because they were contaminated by the ASE noise from the EDFA-amplified pump laser (please also see our answer to reviewer's Comment 5). The spectra shown in Fig. 2c are exactly the comb lines $m=\pm 2, \dots, \pm 11$ both from the transmitter comb C_{Tx} and the receiver comb C_{Rx} , the vacuum (about ± 1.6 nm) in the spectra center is the blocked pump lasers $m=0$ and comb lines $m=\pm 1$.

Second, in Fig. 2a the pilot tone was drawn on the left side of the pump laser because we set the horizontal axis as 'frequency' but forgot to label it. As suggested by the reviewer, in the revised Fig. 2a we have labeled the horizontal-axis as wavelength ' λ ' and moved the pilot tone to the longer wavelength side. Therefore, it is now apparent that the pilot tone is $m=17$ (instead of $m=-17$) and consistent with Fig. 2b and 2c. Thanks again for the meticulous reading of our manuscript and this important correction.

Action taken: In Fig. 2a, the horizontal-axis of the spectrum diagrams have been labeled as wavelength ' λ ', and the pilot tone has been moved to the longer wavelength side, to be consistent with Fig. 2b and 2c.

Comment 4: 4) Related to 3): When looking at the combs in fig 1b it seems like the center part has two dips, one at 1550 (where it is pumped, or..? - and by the way where is the pump located?) and one dip at 1535 or so - why is that? I would like to see a zoomed-in spectrum of the comb lines used in the transmission experiment before the demux, e.g. like in 2c but including also the specific pilot tones and possibly intermediate tones.

Reply: As mentioned in the Method subsection 1, for soliton microcomb generation, we used the auxiliary laser heating (ALH) method to maintain thermal stability of the micro-cavity and to enable reliable access to single-soliton state (details can be found in our prior publications Ref. [29] Light-Science & Applications, 8(1):50, 2019). In the spectra of C_{Tx} and C_{Rx} shown in Fig. 1b, the dips at ~ 1549.9 nm are the blocked pump lasers, and the dips at ~ 1536.2 nm are the blocked auxiliary lasers. The detailed wavelength configuration has been described in the Method subsection 1. Besides, as asked by the reviewer, the zoomed-in spectra of C_{Tx} and C_{Rx} before the demux are shown below.

Figure. The zoomed-in spectra of the C_{Tx} and C_{Rx} comb. The pump lasers (~ 1549.9 nm) and auxiliary lasers (~ 1536.2 nm) both in C_{Tx} and C_{Rx} were blocked using 200 GHz-bandwidth notch filters.

Comment 5: 5) Another issue I have is the dispersive walkoff between the two pilot tones used to seed the Rx comb. In [36] this was discussed and shown to degrade the linewidth of the regenerated comb lines, and since the two seed lines are more widely separated in this experiment than in [36] the dispersive walkoff is larger and potentially more devastating for longer transmission distances. So my question is how the authors' scheme would scale with transmission distance. Apparently it works for 50 km, but would it work for 100/500/1000 kms? This could be addressed in the supplementary.

Reply: We thank the reviewer for this important question.

To answer the reviewer's concern about how our scheme would scale with transmission distance under the influence of dispersive walk-off, we revised the prior phase analysis between the transmitter comb C_{Tx} and the receiver comb C_{Rx} , and explicitly included the effect of fiber dispersion. According to our new analysis, the phase variance of the m -th inter-comb beat note can be wrote as (see detailed derivation in Supplementary Note 1):

$$\begin{aligned}
\langle \Delta \phi_m^{\text{beat}}(t, \tau)^2 \rangle &= \langle [\phi_m^{\text{beat}}(t) - \phi_m^{\text{beat}}(t - \tau)]^2 \rangle \\
&= (A^2 + B^2 + C^2) \cdot \langle \Delta \phi_0(t, \tau)^2 \rangle \\
&\quad + AB \{ -2[\langle \Delta \phi_0(t, T_m - T_d)^2 \rangle + \langle \Delta \phi_{\#}(t, T_m - T_d)^2 \rangle] \\
&\quad + \langle \Delta \phi_0(t, T_m + \tau - T_d)^2 \rangle + \langle \Delta \phi_0(t, \tau + T_d - T_m)^2 \rangle \} \\
&\quad + BC \{ -2[\langle \Delta \phi_0(t, T_d - T_{17})^2 \rangle + \langle \Delta \phi_{\#}(t, T_d - T_{17})^2 \rangle] \\
&\quad + \langle \Delta \phi_0(t, T_d + \tau - T_{17})^2 \rangle + \langle \Delta \phi_0(t, T_{17} + \tau - T_d)^2 \rangle \} \\
&\quad + AC \{ -2[\langle \Delta \phi_0(t, T_m - T_{17})^2 \rangle + \langle \Delta \phi_{\#}(t, T_m - T_{17})^2 \rangle] \\
&\quad + \langle \Delta \phi_0(t, T_m + \tau - T_{17})^2 \rangle + \langle \Delta \phi_0(t, T_{17} + \tau - T_m)^2 \rangle \} \\
&\quad + \left(\frac{m}{17}\right)^2 \langle \Delta \phi_{\text{OPLL}}(t, \tau)^2 \rangle
\end{aligned} \tag{S13}$$

Abundant characteristics can be acquired from Supplementary Eq. S13.

First, as shown in Supplementary Fig. 1a below, it is theoretically confirmed that two-point locking indeed enables substantial suppression of the inter-comb beat note linewidth comparing with the case without two-point locking (namely, when the repetition rate $\omega_{rep}^{\text{Tx}}(t)$ and $\omega_{rep}^{\text{Rx}}(t)$ are treated as independent variables, corresponding to the case: $A = 1 + m\rho$, $B = -1 + m\rho$ and $C = 0$ in Supplementary Eq. S13).

Second, as shown in Supplementary Fig. 1b, when the pump laser linewidth is set to 1 kHz (as the case in our experiment) and the two-point locking scheme is implemented, the effect of dispersive walk-off causes no sizable linewidth broadening to the inter-comb beat notes even if the fiber length is increased to 5000 km. The influence of dispersive walk-off becomes obvious when the fiber length is set to longer than 5000 km and simultaneously the pump laser linewidth is set as 1 MHz, as shown in Supplementary Fig. 1c. To that extent, the dispersive walk-off within the transmission link is not a major limitation to the proposed two-point locking scheme, as long as the transmission distance is not too long and the pump laser has sufficiently low noise. For ultra-long-haul transmission (e.g., >10000 km), however, a lot more issues need to be reconsidered, including the fidelity of two-point locking, link amplification and impairments compensation for the data, which will be investigated in our follow-up studies.

Supplementary Figure 1: Theoretical phase analysis between the transmitter and receiver microcombs. **a.** Comparison of experimental and theoretical inter-comb beat note spectra. **b.** Theoretical inter-comb beat note spectra calculated with 1 kHz pump linewidth. **c.** Theoretical inter-comb beat note spectra calculated with 1 MHz pump linewidth. Parameters used in the theoretical calculation are: $\omega_0 = 193.4$ THz, $k_0 = 6.0 \times 10^6 \text{m}^{-1}$, $k'_0 = 5.0 \times 10^{-9} \text{s} \cdot \text{m}^{-1}$, $k''_0 = 2.5 \times 10^{-26} \text{s}^2 \cdot \text{m}^{-1}$, $T_d = 500$ ns, $\rho = 0.02$.

Finally, it is worth mentioning that the mechanism for soliton microcomb re-generation is fundamentally different from the non-resonant parametric comb re-generation reported in Ref. [36]. Specifically, parametric comb re-generation in Ref. [36] is implemented by cascaded four-wave mixing triggered by two original pilot tones. Fiber dispersion induces phase error between these two original pilot tones, and such phase error will be *inevitably* multiplexed by m times as the m -th parametric comb line is generated, causing considerable broadening of the comb linewidth when m becomes large. In comparison, the re-generation of soliton microcomb C_{Rx} is implemented by a monochromatic pump laser, whose phase coherence is hardly subject to the influence of fiber dispersion, therefore, the linewidth of each comb line within C_{Rx} is not influence by the fiber dispersive walk-off (see Supplementary Eq. S5). In addition, as can be seen from Supplementary Eq. S8, dispersive walk-off gets involved during the process of locking $C_{Rx}(17)$ to the pilot tone $C_{Tx}(17)$. Particularly, when the $C_{Rx}(17)$ is locked to the pilot tone $C_{Tx}(17)$, the other regenerated comb line $C_{Rx}(m)$, $m=1,2,3,4,\dots,16$ are not simultaneously locked to the corresponding transmitter comb line $C_{Tx}(m)$, but subjected to certain time delays T_m due to dispersive walk-off. The overall influence of these time delays to the inter-comb beat note linewidths is determined by Supplementary Eq. S13, as discussed above.

Action taken: As suggested by the reviewer, we have carefully revised the phase analysis to include the effect of dispersive walk-off, and explained how our microcomb re-generation scheme scales with the transmission distance. These contents have been added to Supplementary Note 1.

Comment 6: 6) I have numerous problems with the theory of comb phases in the "methods" section.

a) First of all the purpose of the calculation in eqs (1)-(10) is not clear. What do you want to show? Also I doubt this theory belongs to a methods section. Since this is an experimental paper I expect experimental details, microcomb details, setups and phase locking information, receiver DSP and details on how to measure and show the plotted quantities would appear here.

Reply: We thank the reviewer for this comment. The phase analysis was presented to show that it is theoretically valid to conduct coherence-conserved re-generation of two remote soliton microcombs via the scheme of two-point locking. Also, the derived equations present a guideline for properly choosing the operating parameters (e.g., pump laser linewidth, length of fiber link, OPLL bandwidth, number of comb lines, etc.) so that to apply the coherence-cloned microcombs for more general scenarios in optical data- and tele-communication networks.

As suggested by the reviewer, to keep the article concise and mainly focused on the experimental results, in the revised manuscript we have moved the theoretical phase analysis from the Method section to the Supplementary Note 1.

Action taken: As suggested by the reviewer, we have moved the theoretical phase analysis between microcombs C_{Tx} and C_{Rx} to the Supplementary Note 1.

b) Clearly define the variables and the notation. It is not OK to refer to a specific phase value in time ϕ_{int}^{Tx} as is done after (1) as "intrinsic phase noise (i.e. linewidth)". Rather - I assume - all the ϕ 's are functions of time and a realization of a stochastically varying phase in time, whose statistical properties (which?) are a function of the laser's linewidth. Can you say anything about the statistical properties of these phases? Variance, ergodicity, pdf etc.? This applies also to the Δf_{Rep}^{Tx} which probably varies with time in some unknown fashion, or...?

Reply: Thanks for this important correction. As suggested by the reviewer, in the revised manuscript we have written all the phase terms as functions of time, and clarified the statistical property for the pump laser phase noise (see Supplementary Eq. S15 and S16, as posted below):

variables need to be specified. First, we assume that the pump laser phase noise have a Gaussian probability density function [8]:

$$f(\Delta\phi_0) = \frac{1}{2\pi\sqrt{\Delta f\tau}} \cdot e^{-\frac{\Delta\phi_0^2}{4\pi\Delta f\tau}} \quad (\text{S15})$$

Δf is the Lorentz linewidth of the laser. According to Eq. S15 the phase evolution of the pump laser has a zero mean value and a variance of:

$$\langle \Delta\phi_0(t, \tau)^2 \rangle = 2\pi\Delta f\tau \quad (\text{S16})$$

Besides, as discussed in the Supplementary Note 1, the variation of transmitter comb repetition rate $\Delta f_{\text{Rep}}^{\text{Tx}}$ can be approximately linked to the pump laser noise with a transduction factor ρ (please also see our answer to reviewer's Comment 6(e) below), therefore, the statistic characteristics of $\Delta f_{\text{Rep}}^{\text{Tx}}$ can be derived from the pump laser, as included in Supplementary Eq. S9 and S13.

Action taken: As suggested by the reviewer, the statistical properties for each of the random variables involved in the phase analysis have been clarified. Detailed descriptions have been added in the Supplementary Note 1.

c) The ϕ_{nl} contribution is neglected in the end and seem not to affect the calculation in any way, so I see no purpose in including it. It is also not clear how it would depend on m .

Reply: The nonlinear phase term ϕ_m^{NL} is a practically existing phase perturbation experienced by each comb line as they propagate within the 50 km fiber link, so we believe that it is better to include it in our theoretical model (Supplementary Eq. S1, S2, S4, and S5). Nevertheless, the influence of the nonlinear phase term experienced by each comb line tend to be the same (also see Ref. [S3]), that is, the value of ϕ_m^{NL} does not depend on line index m , thus it disappears when calculating the inter-comb beat note (i.e., frequency and phase difference between different comb lines), as expressed in Supplementary Eq. S6.

d) The authors seem to assume (given the account after eq 10) that the ϕ_{ff} (the phase change due to random fluctuations in the fiber) should also account for the dispersive delay? I think this is not correct and it is not how the delay is modeled. Instead if $\phi^0(t)$ is the phase of line 0 and $\phi^m(t)$ is the phase of line m in to the fiber, the phases after the fiber would be $\phi^0(t-T)$ and $\phi^m(t-T-\Delta t)$, where T is the group delay in the fiber and Δt is the dispersive delay (walkoff) between line 0 and line m . Assuming ergodicity, the common delay T can be neglected but the dispersive delay cannot. This walkoff is a serious problem that contribute to the net phase noise as discussed in Ref [36] and it should be seriously accounted for. The resulting phase noise varies on a time scale related to the laser coherence and not the fiber fluctuations. See also point 5) above.

Reply: We thank the reviewer for pointing out the unclear descriptions regarding the phase changes associated with fiber fluctuation and dispersive walk-off.

First, based on the reviewer's suggestions, we have carefully revised the phase analysis to explicitly include the dispersive walk-off T_m for each comb line, and derived how the inter-comb beat note linewidth evolves with T_m . Moreover, the influence of dispersive walk-off on microcomb re-generation has been compared with the parametric comb re-generation in Ref. [36] (please see our answer to reviewer's Comment 5 above).

Second, the influences of fiber fluctuation and the induced phase error ϕ_{ff} have also been more clearly investigated in the revised phase analysis. Particularly, as shown in Supplementary Eq. S17, the phase error ϕ_{ff} caused by fiber fluctuation is mainly responsible for the low-frequency noise in the inter-comb beat note spectra (<3 kHz, please see the theoretical results in the figure below), which, however, have no essential influence to the high-speed coherent data transmission in our experiment.

Figure. Theoretically calculated inter-comb beat note spectra a. with and b. without random fluctuation of the effective fiber length.

Action taken:

- 1) As suggested by the reviewer, the influences of the dispersive walk-off among different comb lines to the phase coherence between C_{Rx} and C_{Tx} have been carefully analyzed (see details in our answer to reviewer's Comment 5 above).
- 2) The influence of fiber fluctuation induced phase error ϕ_{ff} has also been clarified in the revised Supplementary Note 1, as detailed in Eq. S17 and the corresponding text.

e) Eq (5) seems to assume the variance of the RF contribution of the phase noise increase with line number - which I agree holds for electrooptic combs. However, how the phase noise scales with line number for microcombs is not clear as far as I know. If the authors want to use this model for microresonator-combs they need to support it with a reference or a separate measurement.

Reply: We thank the reviewer for this important comment. In the revised phase analysis, the frequency of the m -th transmitter comb line is written as:

$$f_m^{Tx}(t) = \omega_0(t - T_m) + m \cdot \omega_{rep}^{Tx}(t - T_m)$$

(this equation has been labelled as Supplementary Eq. S2 in the revised manuscript, which is essentially equivalent to the prior Eq. 5 mentioned by the reviewer). Regarding the phase noise, it is seen that $f_m^{\text{Tx}}(t)$ inherits both the pump laser noise (the first term, R.H.S.) and the phase noise originating from the jitter of soliton repetition rate of microcomb C_{Tx} (the second term, R.H.S.), the latter of which increases with line number m . Such “elastic-tape” model is commonly used for Kerr soliton microcombs, supported by Ref. [18, 25, 31] in the main text and Supplementary Ref. [2].

However, it is worth mentioning that (as also pointed out by the reviewer), Supplementary Eq. S2 itself does not guarantee that the overall phase noise of $f_m^{\text{Tx}}(t)$ increases with comb line number m . Particularly, the random jitter of $\omega_{\text{rep}}^{\text{Tx}}(t)$ is usually correlated with the fluctuation of the pump frequency $\omega_0(t)$, which is due to the Raman induced soliton self-frequency shift (SSFS) effect and can be approximately fitted by a linear noise transduction factor $\rho = \partial\omega_{\text{rep}}^{\text{Tx}}/\partial\omega_0$ (see Ref. [18,25,31] in the main text). Then, Eq. S2 becomes: $f_m^{\text{Tx}}(t) = (1 + m\rho) \cdot \omega_0(t - T_m)$, and it is seen that the actual phase noise of f_m^{Tx} can either increase or decrease with the line number m , depending on the sign and actual value of ρ . Such phenomenon has been comprehensively investigated in the recent pre-print paper arXiv:2102.05517v1 (has been cited as Ref. [2] in the Supplementary Information). As can be seen in Fig. 1b of the main text, the soliton microcomb spectra obtained in our experiment exhibit unobvious Raman-induced SSFS, and the ρ factor is measured to be relatively small: 0.02, which agrees reasonably well with previously measured values: 0.022 in Ref. [25], 0.025 in Ref. [31], and 0.067 in arXiv:2102.05517v1.

Besides, after two-point locking, the phase noises of the inter-comb beat notes depend on a multitude of variables including the dispersive walk-off T_m , the apparatus-induced time delay T_d , the phase noise of the pump laser, and the residual noise of the OPLL. These combinational effects together generate an *intertwined* relationship between the inter-comb beat note noise and the line index m , as expressed in Eq. S13 and illustrated in Supplementary Fig. 1.

We hope our explanations can answer the reviewer’s questions, and we will be more than happy to provide further information if needed. Thank you again for these excellent questions and suggestions.

Action taken: Discussion about how the inter-comb beat note phase noise change with comb line number m has been presented to the Supplementary Note 1, please see Supplementary Eq. S9 and the corresponding text.

Comment 7: 7) More details and explanations are needed on the coherent receiver DSP. I suggest providing a detailed explanation in a supplementary chapter. Specifically:

Reply: As suggested by the reviewer, more detailed explanations about the receiver DSP algorithms have been added to the Method subsection 3. Besides, the specific questions raised by the reviewer have been answered point-by-point below.

Action taken: More detailed explanation about the receiver DSP algorithms has been added to the Method subsection 3 (page 12 and 13 of the main text).

-the Volterra equalizer is nonstandard. What does it do? Why is it needed? What is the performance without it?

Reply: The Volterra equalizer is an algorithm-block in our coherent receiving program kit that aims at dealing with the nonlinear distortion of the received data signal. However, the nonlinear distortion was not prominent in our 50 km transmission experiment, therefore, the actually adopted Volterra equalizer mainly consists of 30 linear filter taps, with only 2 second-order taps and 1 third-order tap. As shown in the figure below, the performance of Volterra equalizer has no essential difference than that of a linear FIR equalizer.

Figure. Comparison of data receiving performance using Volterra equalizer and linear equalizer, exhibiting no essential difference. The data are from CH 1 carried by coherence-cloned microcombs.

- is the received data pol muxed or single polarization, and what poldemux algorithm is used?

Reply: Our experiment focused on investigating the simplified FOE and CPE operations enabled by the outstanding frequency and phase coherence between the two-point locked microcombs, so we only used single polarization data modulation, to avoid extra perturbation from the polarization-demultiplexing process. Nevertheless, the demonstrated benefits regarding FOE and CPE enabled by microcombs also apply to dual-pol data transmission. In fact, the ultra-low phase noise between the coherence-cloned microcombs holds potential to minimize the influence of equalizer enhanced phase noise (EEPN) and in turn facilitate accurate demultiplexing of dual-pol data, which will be investigated in our following studies.

- apparently pilot-aided CPE is used. Details please - why is it needed if the LO is 'cloned' to the transmitter source? Note that this is a critical point because the whole purpose of this paper falls if the data cannot be retrieved with a blind CPE. Does it work without the CPE? What is the penalty without the CPE? What is the rep rate for the pilot symbols and what format are they?

Reply: We thank the reviewer for the specific questions.

First and importantly, blind CPE algorithms (such as blind phase search BPS) can also be used to retrieve the data in our experiment, just at the cost of heavier computation than pilot-aided CPE. We have added the results of coherent data receiving using BPS-based CPE in Supplementary Note 3. As shown in Supplementary Fig. 4a and 4b, when blind CPE is used (BPS with block size $N_{BPS}=32$ samples and test phase number $B=32$), the BER curves remain essentially identical as obtained by the pilot-aided CPE.

Supplementary Figure 4: Coherent data performance of different CPE algorithms. **a.** Measured BER as a function of BPS-based CPE rate. **b.** Measured BER as a function of pilot-aided CPE rate. **c.** Measured BER using decision-directed adaptive equalizer with or without CPE in the loop. The DD-LMS equalizer contains 17 taps and the step size is 0.0005. The taps are pre-converged using 500 training symbols, and then the equalizer is switched to the decision-directed mode to trace the slow phase drift between C_{Tx} and C_{Rx} .

Second, regarding the question ‘Does it work without CPE’, the answer is yes, but adaptive equalizer with sufficiently high performance is needed to compensate the residual phase drifts between the coherence-cloned microcombs. As shown in Supplementary Fig. 4c above, when DD-LMS adaptive equalizer is implemented to retrieve the 16-QAM data collected in our experiment, the data receiving BER can be controlled well below the desired threshold from CH1 to CH10, without conducting CPE for any of the channels. However, when coherence-cloned microcombs are adopted in a realistic network, either the adaptive equalizer should replace or work collaboratively with the CPE module depends on the specific system parameters (such as the fiber link length, channel impairments, polarization sensitivity, cost and power budget, etc.), which we believe is an important topic to follow up.

Third, regarding the question ‘What is the rep rate for the pilot symbols and what format are they’, the pilot symbol repetition rate was altered to investigate the phase stability of different channels. As shown in Fig. 3c, the horizontal axis is labelled as ‘skipped block’, which is equal to the reciprocal of the pilot repetition rate. For example, ‘1000 skipped blocks’ means that the pilot symbol block is sent once every 1000 blocks (each block contains 32 symbols). The format of the pilot symbols is single polarization 16-QAM, same as the payload symbols.

Finally, it is worth mentioning that our scheme should be distinguished from the analogue coherent receiver architecture (see Journal of Lightwave Technology, vol. 35, no. 21, 2017, pp. 4650–4662), for which phase estimator and OPLL are embedded INSIDE the coherent receiver for analogue phase locking, thus no digital CPE is needed. On the other hand, as analogue receivers are revisited recently as a potential solution for low-power data-center interconnect system, the phase stability of coherence-cloned microcombs holds potential to relax the OPLL bandwidth and circuits complexity of an analogue coherent receiver, which would be another interesting topic to consider.

Action taken:

- 1) As suggested by the reviewer, the data receiving performance using blind CPE (i.e., BPS) has been added in Supplementary Fig. 4a.
- 2) Coherent data receiving using adaptive equalizer have been presented in Supplementary Fig. 4c, demonstrating that the coherence-cloned microcombs can work without CPE, if the equalizer module is able to continuously compensate the residual phase error between C_{Tx} and C_{Rx} .

8) You seem to compare the two-point locking to the free-running case, but what about 1-point locking - i.e. if you use line 0 to create the Rx comb and skip line 17 - is that case studied at all, and if so what is the performance?

Reply: We thank the reviewer for this question. In fact, the '1-point locking' scheme had already been investigated in our experiment. As presented in Fig. 2f, Fig. 2g, Fig. 3b and Fig. 3c, the data curves labelled as 'w/o two-point locking' were in fact obtained using '1-point locking' scheme, that is, the conveyed pump laser was used to generate the receiver comb C_{Rx} , but $C_{Rx}(17)$ was not locked to the pilot tone $C_{Tx}(17)$. Besides, the data curves labelled as 'independent carrier and LO' in Fig. 3c means the 'free-running case', where the carrier and LO were from independent lasers. As suggested by the reviewer, we have clarified this in the caption of Fig. 1.

In general, as can be seen in Fig. 3b and 3c, '1-point locked' microcombs have inferior frequency and phase stability than two-point locked microcombs, therefore they require more frequently updated CPE to achieve the same BER.

Action taken: The following clarification has been added to the revised manuscript.

- 1) Page 4 paragraph 1: "At this stage, C_{Tx} and C_{Rx} can be considered as being 'one-point locked:..."
- 2) Fig. 1c caption: "Without two-point locking means that the conveyed pump laser $C_{Tx}(0)$ is used to generate the receiver microcomb C_{Rx} , but $C_{Rx}(17)$ is not locked to the pilot tone $C_{Tx}(17)$."

These are some of the questions I would like to see addressed satisfactory before I would approve publication of this submission.

Reply: Above we have addressed the reviewer's comments point-by-point, and hopefully our explanations can answer your questions clearly. We will be more than happy to provide more information if needed. Thank you again for these excellent comments which really help us to improve our work.

Reviewer #3 (Remarks to the Author):

Comment 0: The manuscript by Yong Geng et al. entitled "Coherent optical communications using coherence-cloned Kerr soliton microcombs" presents the experiments dedicated to coherence-cloned regeneration of DKS microcombs and their exploitation as the carriers and local oscillators for terabit coherent interconnect. The microcomb on the receiver side is regenerated by means of a microresonator which has a free spectral range similar to that of the transmitter, located 50 km away, by the conveyed pump that is transferred together with the channels for information transfer. In order to further increase the coherence between the microcombs, a two-point locking technique of the 17th comb lines is employed, so that beat note linewidth drops from > 3 kHz down to ~ 5 Hz. High coherence rate enables simpler and less power-consuming digital signal processing (DSP) for frequency offset estimation (FOE) and carrier phase estimation (CPE) to retrieve the information.

Reply: We thank the reviewer for reviewing our manuscript and for the positive comments about our work. Below we have addressed your comments point-by-point.

Comment 1: The manuscript is written well. The abstract contains a concise overview of the paper main results as well as their place in the framework of the considered field of research. Although the general distribution of the material between the main text and the supplementary information makes sense, the latter could be still enriched with explicit description of the conventional FOE and CPE algorithms that are said to be simplified.

Reply: We thank the reviewer for this suggestion. In the revised manuscript we have added a section in the Supplementary Information that explicitly describes the conventional FOE and CPE algorithms, and compares the complexity of these FOE and CPE algorithms with or without using coherence-cloned microcombs (please also see our answer to reviewer's Comment 4 below). Thanks.

Action taken: As suggested by the reviewer, detailed descriptions about FOE and CPE algorithms have been added to the Supplementary Note 3.

Comment 2: The work is consistent with previous works on coherent data detection where the microcombs are used as a multiwavelength source that can be used instead of independent laser arrays to carry information on long distances and to be coherently retrieved on the receiver side. In our view, the introduction is properly composed so that it clearly describes the field and reasonably motivates the current research work. Although the coherent detection of the carrier microcomb by a local oscillator microcomb as well as regeneration of microcombs by a comb line pumping a remote microresonator were previously done in Ref. [MarinPalomo2017] and Ref. [Liao2019], correspondingly, the technique of two point locking, which principal foundations were described in Ref. [Fortier2011], was for the first time used to increase the coherence between the carrier and regenerated combs. On the other hand, as it has been demonstrated in Ref. [Jang2018], it is possible to generate a synchronized microcombs when the slave microcomb is locked to the master microcomb. Would it be possible to perform microcombs synchronization in the experiments considered in the

manuscript instead of two-point-locking and pump conveying?

Reply: We thank the reviewer for this comment. However, the scheme of all-optical synchronization of two soliton microcombs demonstrated in Ref. [Jang2018] is not perfectly suitable for the application of coherent optical communication, considering the following two issues:

First, all-optical locking of two soliton microcombs usually has small locking range, for example, for silicon nitride micro-ring resonators with a Q-factor of about 2 million, the locking range is only $< \pm 3$ MHz (see Fig. 3 in Ref. [Jang2018]). This requires extremely precise control of the resonators' size (i.e., the FSR) and quite difficult to realize in the process of high-yield chip production.

Second and more importantly, for all-optical synchronization of two soliton microcombs, the majority of the master comb lines need to be used to form a sufficiently short synchronizing-pulse within the slave micro-cavity, which means that lots of the transmitter comb lines cannot be used to carry data information, making the scheme unacceptable in term of spectrum efficiency.

Comment 3: The issue of possible spectral purity degradation of the pump and pilot signals translated to the receiver due to the effect of nonlinear cross-phase modulation is addressed and found to be non-essential in the corresponding experimental comparison between the two cases when the both signals propagate together with the data carriers and without them. The explanation given to this observation is that fiber dispersion induces the spatiotemporal walk-off among signals at different wavelength channels and is confirmed by numeric simulations. Although the parameters set for these simulations are indicated, it could be more educational to include the basic equations comprising the numerical model and the method used for its simulations.

Reply: We thank the reviewer for this suggestion. The basic equation for the simulation results mentioned by the reviewer is the nonlinear Schrödinger equation (NLSE) that models data signal transmission in an optical fiber:

$$\frac{\partial E}{\partial z} = -\frac{1}{2}\alpha E - i\frac{\beta_2}{2}\frac{\partial^2 E}{\partial T^2} + \frac{\beta_3}{6}\frac{\partial^3 E}{\partial T^3} + i\gamma|E|^2E - \frac{\gamma}{\omega_0}\frac{\partial}{\partial T}|E|^2E$$

Herein $\beta_2 = -20 \times 10^{-27} (\text{s}^2/\text{m})$ and $\beta_3 = 1.5 \times 10^{-40} (\text{s}^3/\text{m})$ is the second- and third-order fiber dispersion respectively; $\alpha = 0.2$ dB/km is the fiber loss coefficient, $\gamma = 1.0 \text{ W}^{-1} \cdot \text{km}^{-1}$ is the fiber nonlinear coefficient; the last term stands for self-steepening effect; Raman effect is excluded in the NLSE considering that the relevant laser wavelengths in our simulation are all within the C-band. The NLSE is numerically solved via the split-step Fourier method using commercial simulation software kit (Optisystem).

Action taken: As suggested by the reviewer, the basic NLSE for our simulation has been introduced in the Supplementary Note 2.

Comment 4: Rigorous analytic description of coherence analysis between the transmitter and receiver combs provides a clear insight into the origin of phase correlations between the channels observed in the experiment. This analysis provides a reasonable justification of the DSP algorithms covered in the paper. For more strict assertion of what advantages the proposed detection scheme

offers, it would be useful to evaluate the asymptotic value of operations' number that are required by the conventional realizations of FOE and CPE and their upgraded counterparts.

Reply: We thank the reviewer for this nice suggestion. In the revised manuscript, we have evaluated the asymptotic values of operation number required by conventional FOE and CPE algorithms, and presented a quantitative analysis about the saving of DSP operations regarding FOE and CPE enabled by the coherence-cloned microcombs.

Action taken: As suggested by the reviewer, a quantitative analysis about the DSP operations required by FOE and CPE algorithms has been presented in Supplementary Note 3. The main results are summarized in the Table below.

Digital operations required by typical FOE and CPE algorithms			
	Real Multiplications	Real Additions	Comparisons
FOE (4-th FFT)	$(2N_{\text{FOE}}\log_2(N_{\text{FOE}}) + 10N_{\text{FOE}} + 2)$	$(3N_{\text{FOE}}\log_2(N_{\text{FOE}}) + 5N_{\text{FOE}})$	N_{FOE}
BPS-based CPE	$6BN_{\text{BPS}} + 4N_{\text{BPS}}$	$6BN_{\text{BPS}} - B + 2 + 2N_{\text{BPS}}$	B
Pilot-aided CPE	$8N_{\text{pilot}}$	$4N_{\text{pilot}}$	N_{pilot}

Comment 5: In Fig. 2b, SNR and BER data are given, however, it is not clear which case these data correspond to: either it is for the two-point locked local oscillator (LO) comb or for the comb without it. Anyway, could you compare the both cases with the case when there is no cloning of the LO comb, please, to properly outline advantages offered by this technique?

Reply: We thank the reviewer for this comment.

First, in Fig. 2b, the presented SNR and BER results were obtained with two-point locked microcombs. This has been clarified in the revised Fig. 2b caption.

Second, we thank the reviewer's suggestion to compare the data receiving performance enabled by two-point locked microcombs with the case when there is no cloning of the LO. As can be seen in the figure below, when there is no cloning of the LO comb, the SNR and BER of the detected data become much worse than achieved by phase cloned LO comb. NOTE that the results in this figure are obtained by conducting only one CPE for the whole data sequence, so the BER values reflect the general phase stability of each channel with or without phase cloning. More detailed information about the data transmission performances with and without phase cloning microcombs has already been presented in Fig. 3b and 3c in the main text, wherein the curves labeled as 'w/ two-point locking' corresponding to phase cloned LO comb; while the curves labeled as 'w/o two-point locking' corresponding to LO comb without phase cloning.

Figure. Comparison of data receiving performances enabled by microcombs (a) with and (b) without phase cloning. Note, that only one CPE is used for the whole data sequence to obtain the SNR and BER, so the BER values reflect the general phase stability of each channel with or without phase cloning.

Action taken:

i). The caption of Fig. 2b has been modified to: "b. SNR and BER measurements for all the 20 data channels, using two-point locked microcomb lines as carriers and LOs."

Comment 6: In Fig. 3b, the phase evolution is given for the scale of tens of microseconds. At the same time, for the given data transmission rate, a time slot duration for one symbol transmission is on the order of tens of picoseconds. On this time-scale, the considered large-time-scale phase evolution does not contribute too much to the decoding error as the phase fluctuations are small. Why does one have to use two-point locking to increase the stability on the larger time scale, if that is the case?

Reply: We thank the reviewer for this question, but we respectfully disagree with your idea.

If carrier phase estimation (CPE) is conducted for every data symbol, then there is no need to care about the carrier-LO phase stability longer than the time slot of one symbol. However, for coherent data receiving, CPE processes are usually complicated and power hungry, therefore, conducting CPE for each data symbol is impractical in terms of system complexity and energy efficiency.

Practically, it is desired that the CPE is conducted at an as-low-as-possible rate (much slower than the symbol rate), which in turn requires as-high-as-possible phase stability between the data carrier and LO. This is the main purpose to generate and use coherence-cloned microcombs.

As shown in Fig. 3c, for example, using phase cloned microcombs as the data carrier and LO, the large-time-scale phase evolution for CH 1 is quite stable across the entire data sequence (Fig. 3b top panel red-curve), therefore only 1 CPE is enough for receiving the whole 400,000 symbols with satisfactorily low BER (Fig. 3c green solid curve). In comparison, when we use microcombs without phase cloning, the large-time-scale phase evolution for CH 1 exhibited higher fluctuation (Fig. 3b top panel blue-curve), and our measurement shows that CPE update is needed every 32,000 symbols (i.e., 1000 skipped blocks, see Fig. 3c green dashed curve) to reach the same BER, which correspondingly consumes 13 times more CPE operations and energy than using phase cloned microcombs.

Comment 7: In summary, the experiments is a system level experiment that demonstrates coherence cloning and is a very useful addition to the application of microcombs in coherent communication. The manuscript and experiments are of high quality.

Reply: We thank the reviewer for the positive comment about our work.

Comment 8: Please find below a set of other questions and technical remarks which would improve the quality of the manuscript:

Reply: We thank the reviewer for these detailed technique remarks and corrections, which have been addressed point-by-point in the revised manuscript, as shown below.

Comment 8a: Line 135: BER stands for bit-error rate, however, in Ref [MarinPalomo2017] it stands for bit error ratio that is a more correct way to express the meaning of this quantity as it is dimensionless, whereas rate usually represents the speed of a physical measure evolution over time.

Reply: Thanks for this important correction. Although BER is frequently referred to as "bit error rate", we now realized that it is more accurate to define it as "bit error ratio" considering its actual meaning (https://blogs.keysight.com/blogs/tech.entry.html/2019/03/10/ber_is_it_bit_erro-PIIE.html).

Action taken: As suggested by the reviewer, "bit error rate" has been corrected as "bit error ratio" in the revised manuscript.

Comment 8b: Line 281: ... linearly depends ...

Action taken: This grammatical mistake 'is linearly depends' has been corrected.

Comment 8c: Fig. 1b caption: ... must be implemented to recover ...

Action taken: The typo 'recovery' has been corrected into 'recover'.

Comment 8d: Fig. 2c: it is not clear how OCNR was evaluated. What was taken as the noise floor? Resolution bandwidth should also be added to properly represent the data collected from an optical spectrum analyzer.

Reply: Thanks for this detailed question. In Fig. 2c the OCNR is evaluated as the ratio between the optical power of each comb line to the power of adjacent ASE floor. As suggested by the reviewer, we have added a double-headed line to mark how OCNR is evaluated in Fig. 2c, and the 0.02 nm resolution bandwidth has been added to the inset of Fig. 2c.

Action taken: A double-headed line has been added in Fig. 2c to illustrate how OCNR is evaluated, and the 0.02 nm resolution bandwidth has been labelled.

Comment 8e: Fig. 3c: BER should not be ranged up to 10. The ratios of more than 0.1 are meaningless to indicate.

Reply: Thanks for this correction. The BER axis scale was set up to 10 in order to have more space to put the legends in Fig. 3c. As suggested by the reviewer, we have deleted the BER labels above 0.1.

Action taken: The BER labels above 0.1 have been deleted in Fig. 3c.

Comment 8f: Fig. 3f: why the shapes comprising the constellations are tilted? Why these shapes' distribution is non-uniform? (compare with Fig. 2g in Ref. [MarinPalomo2017])

Reply: We thank the reviewer for this detailed question. The slight distortion of the constellation diagrams shown in Fig. 3f is mainly attributed to the nonideality of our IQ-modulator (such as non-linear amplification of the electrical signals and imperfect bias voltage). As shown below, the back-to-back (B2B) 16-QAM signal already exhibits mildly stretched constellation diagrams. After 50 km fiber transmission, the original constellation distortions are slightly enhanced. However, as long as the BER values are below the desired threshold (i.e., 3.8×10^{-3} for Fig. 3c and 3f), such minor constellation tilts do not influence the validity of our investigation presented in Fig. 3.

Comment 8g: Fig. S1b: ... between the models including and excluding the XPM effect ...

Action taken: As suggested by the reviewer, the phrase "between the cases with and without XPM effect" has been changed to "between the models including and excluding the XPM effect" in Supplementary Fig. 2b

Comment 8h: Fig. S2a: between OBPF and the circulator on the output there is a part of the setup scheme with the oval shape which role is not clear.

Reply: The oval pattern is a power splitter that splits the receiver microcomb C_{Rx} into two parts: one part serves as the LO laser array delivered to the coherent receiver, and the other part is used to provide $C_{Rx}(17)$ for two-point locking.

Action taken: The oval pattern has been labelled as "Power splitter" in Supplementary Fig. 5a.

REVIEWERS' COMMENTS

Reviewer #1 (Remarks to the Author):

I think the authors did a good job in addressing my comments. The reply is satisfactory, and I now support publication of the manuscript.

■ **Reviewer #1 (Remarks to the Author):**

I think the authors did a good job in addressing my comments. The reply is satisfactory, and I now support publication of the manuscript.

Reply: We thank the reviewer for his/her appreciation for our efforts and the recommendation for publication.